# Reducing Symbiosis Bias Through Better A/B Tests of Recommendation Algorithms

## ABSTRACT

It is increasingly common in digital environments to use A/B tests to compare the performance of recommendation algorithms. However, such experiments often violate the stable unit treatment value assumption (SUTVA), particularly SUTVA's "no hidden treatments" assumption, due to the shared data between algorithms being compared. This results in a novel form of bias, which we term "symbiosis bias," where the performance of each algorithm is influenced by the training data generated by its competitor. In this paper, we investigate three experimental designs–cluster-randomized, data-diverted, and user-corpus co-diverted experiments–aimed at mitigating symbiosis bias. We present a theoretical model of symbiosis bias and simulate the impact of each design in dynamic recommendation environments. Our results show that while each design reduces symbiosis bias to some extent, they also introduce new challenges, such as reduced training data in data-diverted experiments. We further validate the existence of symbiosis bias using data from a large-scale A/B test conducted on a global recommender system, demonstrating that symbiosis bias affects treatment effect estimates in the field. Our findings provide actionable insights for researchers and practitioners seeking to design experiments that accurately capture algorithmic performance without bias in treatment effect estimates introduced by shared data.

## CCS CONCEPTS

• **Information systems** → **Personalization**; *Collaborative filtering*; • **Computing methodologies** → *Modeling and simulation*; • **Mathematics of computing** → **Probability and statistics**.

## KEYWORDS

Interference, A/B testing, Recommendation Algorithms, Symbiosis Bias, Experiment Design

**ACM Reference Format:**
Anonymous Author(s). 2025. Reducing Symbiosis Bias Through Better A/B Tests of Recommendation Algorithms. In *Proceedings of Proceedings of the ACM Web Conference 2024 (WWW '25) (WWW '25)*. ACM, New York, NY, USA, 14 pages. https://doi.org/XXXXXXX.XXXXXXX

## 1 INTRODUCTION

As algorithmic recommendations and personalization have become increasingly ubiquitous in digital settings, it has become more and more common for both researchers and practitioners to conduct randomized experiments in which the "control" and "treatment" interventions that are being compared are algorithmic in nature. For instance, A/B tests are used to compare the efficacy of algorithms for recommending content [22], suggesting social media contacts [41], and setting prices [1]. Obtaining unbiased treatment estimates from simple, user-randomized experiments relies on the stable unit treatment value assumption (SUTVA) [40, 42], which states both that there is no interference between units, and that there are no "hidden treatments." However, many experiments that compare the efficacy of different algorithms violate SUTVA's no hidden treatments assumption, due to the fact that each unit's potential outcomes are a function not just of the algorithm that it is exposed to, but also of the data that the algorithm has been trained on. In many cases, algorithmic recommendations are updated over the course of an experiment using newly-generated training data, and no distinction is made between data that is produced by subjects exposed to different treatment interventions. As a result, simple experiments meant to evaluate the impact of different algorithmic interventions are often not able to observe the actual counterfactual quantities of interest. We refer to bias in Total Treatment Effect (TTE) estimates arising due to this phenomenon as "symbiosis bias."

In this paper, we use theory and simulation to investigate the efficacy of three different experiment designs at reducing symbiosis bias in experimental comparisons of recommendation algorithms:

- **Cluster-randomized experiments:** In cluster-randomized experiments, users are first assigned to different clusters (typically based on historical actions and/or other observables). Treatment is then randomized at the cluster-level, as opposed to the user-level.
- **Data-diverted experiments:** In data-diverted experiments, treatment is randomized at the user-level. However, during the experiment, the treatment (control) algorithm is only updated with data produced by users assigned to the treatment (control).
- **User-corpus co-diverted experiments:** In user-corpus co-diverted experiments, both users and items are randomly assigned to treatment or control. During the experiment, treatment (control) users are only able to see/interact with treatment (control) items.

Insofar as symbiosis bias arises from the treatment (control) algorithm having access to training data that would not have existed under the counterfactual where all units were exposed to the treatment (control), the three experiment designs are similar in that they aim to restrict the pool of data available to each algorithm to that which would be available under the relevant counterfactual.

Both our theoretical model and our simulation framework establish a number of important facts that can guide both practitioners and researchers when assessing if and how to address symbiosis bias through experiment design. First, symbiosis bias does exist

in experimental comparisons of recommender algorithms, and the severity of symbiosis bias depends on factors, including the particular set of algorithms being. Second, symbiosis bias is often asymmetric, i.e., in a comparison of two algorithms, one algorithm benefits more from data produced by its competitor than vice versa. Finally, while designs like data-diverted experimentation has the potential to reduce symbiosis bias, such experiment designs can introduce their own type of bias stemming from the fact that for many algorithms, performance degrades with access to less training data and/or a smaller corpus of items to recommend.

We also contribute to the emerging research literature on symbiosis bias by demonstrating the existence of this novel form of bias (and the efficacy of cluster randomization at reducing said bias) using data from a country-randomized A/B test conducted on a large industrial recommender system. In the experiment, the treatment algorithm boosted recently published content relative to the control algorithm. Our analysis shows that control countries with higher pre-experiment content overlap with treated countries had higher engagement with boosted content during the experiment. This suggests that exploration data generated by users in treated countries influenced recommendations to users in control countries, thus introducing symbiosis bias into the experiment.

Our results extend a number of recent papers focused on bias arising from data-related spillovers, including Musgrave et al. [38], Goli et al. [17], and Si [49]. While each of these papers make valuable contributions, they also all have important limitations. For instance, Musgrave et al. [38] considers data-related spillovers in settings where users issue search queries (e.g., web search) and proposes query-randomized experiments. However, not all products that issue algorithmic content recommendations involve explicit queries from the user. Goli et al. [17] proposes a bias correction approach for ranking experiments, but makes a number of strong assumptions (e.g., that demand for each item is independent the ranking of other items) and focuses exclusively on the steady state reached after repeated interactions between algorithms. The weighted training approach proposed by Si [49] requires the experiment designer to estimate the probability that each item will be recommended in both the control and treatment, which may be difficult to do in practice. In contrast, we evaluate the efficacy of multiple different parsimonious experiment designs that require minimal assumptions, can be used in cases where recommender systems have not yet reached equilibrium, and do not require users to, for example, issue search queries.

The remainder of this article is organized as follows. In Section 2, we review the related research literature on A/B testing in the presence of feedback loops. Section 3 puts forth two potential outcome models: a dynamic potential outcomes model, which extends Neyman's finite population causal model to depend not just on the treatment assignment, but also on the available data, and a more tractable equilibrium model, under which we can get intuition for the amount of symbiosis bias present under different experiment designs. Section 4 uses a simulation framework to explore the amount of symbiosis bias present under different experiment designs in a dynamic setting. In section 5, we analyze data from a field experiment conducted on a production scale recommender system used by millions of users, and identify evidence of symbiosis bias. Finally, Section 6 concludes.

## 2 RELATED LITERATURE

Recommendation algorithms power online platforms used daily by billions of people, and there exists an extensive literature [36, 44, 46, 51, 59, 61, 63] studying the biases that arise during their evaluation, covered recently in a comprehensive survey by Chen et al. [10]. Of these biases, we focus our efforts on the "symbiosis bias" that arises from two or more recommendation algorithms sharing data in a live experiment. This focus on comparisons made in live experiments distinguishes our work from previous investigations into the biases that plague offline evaluation methods [26, 28], from investigations into the biases of an algorithm's past recommendation on its future self [32, 50, 53], or from investigations that explore the ecosystem impact of feedback loops, separate from A/B testing concerns [9]. We adopt the perspective that feedback loops *within* a given algorithm should be measured by an A/B test, while feedback loops *between* two algorithms are a source of bias and should be suppressed. Accurate TTE measurement thus requires breaking some feedback loops and not others, which distinguishes it from related works.

A/B testing is the statistical foundation of causal inference on web systems [33], providing valid causal inference under the stable unit treatment value assumption (SUTVA) [42]. This assumption is violated when the outcome of one unit depends on the treatment assignment of another unit and/or there are "hidden treatments." When two arms of an experiment share a common data pool, both of SUTVA's requirements fail to hold. More generally, the challenges of A/B testing under feedback loops have been recognized across a variety of web applications including ad placement systems at Microsoft [6], recommender systems at Netflix [54] and Google [11, 52], and ranking systems at Meta [20], LinkedIn [39] and Tencent [62].

A variety of solutions have been proposed to address the problem of symbiosis bias, some of which are *experiment design-based* and some of which are *analysis-based*. In this paper, we use theory and simulation to compare three different design-based approaches to reducing symbiosis bias: data-diversion, user-corpus co-diversion, and cluster randomization.[1]

Perhaps the most straightforward design-based approach is **data-diversion**, in which each algorithm trains on only its own data. Data-diverted experiments are considered by several prior works [19, 27, 49, 52, 58]: Jeunen [27] (under the name *data-siloed*) cites the increase in variance from using less data as a barrier to implementing data-diverted experiments in practice. Si [49] identifies the data-inefficiency of the approach as a barrier to adoption. Our work identifies a third reason to be wary of data-diverted experiments: using less data to train a recommendation model degrades the performance of each model, but this degradation may be unequal across the two arms, and especially so if the two arms are unequally sized. This introduces a new bias into the treatment effect estimate. The

---

[1]Another related experiment design is the switchback design [5], which attempts to mitigate network interference by alternating between two states: all units in treatment and all units in control. Switchback designs incur no interference between units during a given time period but suffer from carryover effects [25] between experimental periods. Glynn et al. [16] use switchback experiments to measure feedback loops in data collection for inventory constrained Markov Decision Processes under the name "temporal interference," further studied by Farias et al. [14]. In practice, these solutions require large amounts of experimental traffic for short amounts of time. This makes them unpalatable in practical settings similar to the one presented in Section 5.

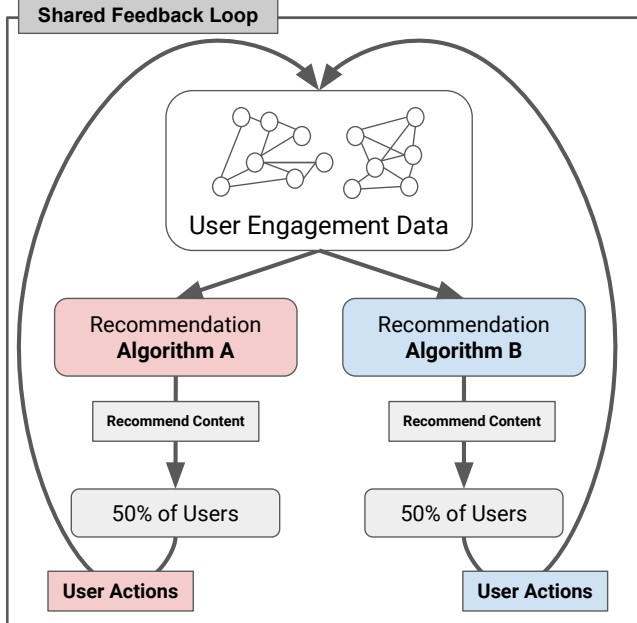

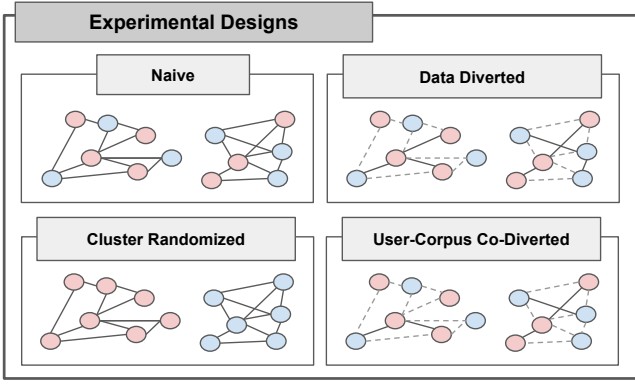

**Figure 1: Four experimental designs for A/B tests of recommendation algorithms. User engagement data is depicted by a graph whose nodes represent users and edges represent the extent to which one user's engagement data informs another user's recommendations. In the naive experiment all data is used to train both models. In the data-diverted experiment data from one arm is not used to train the other; this is represented by the removal of edges in the graph, reducing the information available to make recommendations in each arm. In the cluster-randomized experiment the users are first clustered into $K$ clusters ($K = 2$ shown here) and then clusters are randomized to treatment or control; if data is perfectly clustered this method removes bias-inducing cross-cluster edges without degrading recommendations by forcibly removing edges. In the user-corpus co-diverted experiment, both users and items are randomized into treatment and control, and treatment (control) users can only consume treatment (control) items, causing even more edges between users to disappear relative to the data-diverted design.**

problem of unequal sizes can be mitigated by splitting the data into three parts: two equally-sized treated and control arms, and one non-involved arm that receives the control but does not participate in the analysis [52]. Wu et al. [58] propose a form of data-diversion called the "fair bucket" design to estimate the long-term benefit of exploration using a short-term experiment.

Another design-based approach is the **user-corpus co-diverted experiment** [11, 52]. This approach refines the data-diverted experiment using a multiple randomization design [2, 31] that presents a randomized subset of content to treated users and another randomized subset to control users. This work gets around the bias inherent in data-diverted experiments by assuming the existence of a scaling law by which the system's behavior is reliably predicted from random corpus ablations.

A third design-based approach that has not previously been considered in the literature on symbiosis is the **cluster-randomized experiment** [56]. Cluster-randomized experiments first cluster users according to some similarity graph, then randomize all units in each cluster to either treatment or control. Cluster randomization reduces interference when the graph accurately captures the mechanism of interference by ensuring that units are influenced by other units with their same treatment assignment. We apply the one-sided bipartite experiment framework of [7], which captures A/B testing on recommendation systems represented by a bipartite graph between users and items.

In addition to these design-based approaches, there are also multiple analysis-based approaches to reducing symbiosis bias. The **weighted training approach** [49] modifies the loss function of each recommendation algorithm to downweight data collected by the alter arm if it was unlikely to be collected by the ego arm. This prevents the ego from free-riding on exploration performed by the alter. The weighted training approach requires the platform to estimate the probability that each arm produces each recommendation. It is similar in spirit to off-policy evaluation, including using random exploration to debias offline evaluation as suggested by Jadidinejad et al. [26]. Another approach is to **model the interference mechanism** and correct for symbiosis bias with a modified estimator. Zhan et al. [60] propose the "recommender choice model" of item recommendation in a creator-randomized experiment on a video-sharing platform. Under this model, whose parameters can be estimated from known item features and a known score given by each algorithm to each item, the authors derive a doubly-robust estimator of the TATE.

Beyond the literature focused specifically on symbiosis bias, there is also a growing body of research studying the closely related problem of A/B testing under feedback loops in ranking systems. Musgrave et al. [38] study a ranking system in which treatment and control contribute to shared features; they identify how to leverage the search queries in a query-diverted experiment that isolates these feedback loops. Another approach to mitigate interference in ranking experiments is the counterfactual interleaving design [20, 39, 62], in which both the treatment and control arms produce a ranked list of recommendations and the design merges the two lists in a way that minimizes interference. The counterfactual interleaving design is not applicable to recommender systems in which only one item is recommended. Finally, Goli et al. [17] develop a method

to de-bias ranking algorithms subject to interference by identifying items which are ranked near their counterfactual ranking.

In this paper, we contribute to the growing literature on symbiosis bias by providing both theory- and simulation-based comparisons of three different design-based approaches to reducing this bias, including one design (cluster randomization) that has not previously been considered in the context of symbiosis bias. Furthermore, we use data from a large-scale field experiment to both document the existence of symbiosis bias in the field and provide a preliminary evaluation of the efficacy of cluster randomization at reducing said bias.

## 3 A THEORETICAL MODEL

Large recommender systems are in practice built from many interconnected components. The complexity and specificity of each implementation means that data scientists who study the performance of these systems often treat them as a "black box". To formally define symbiosis bias, we briefly introduce a time-indexed model in potential outcome notation. Because this dynamic model is largely intractable without further assumptions, we then propose an equilibrium model, which we find more practical for capturing the intuition of different solutions to the symbiosis bias problem.

We follow an extension of Neyman's finite population causal model in which the outcome of interest for unit $i$ at time $t$ is a function of just unit $i$'s (time-invariant) treatment assignment $z_i$, but also of the training data available at time $t$, $d_t$.

$$\mathbf{Y}_t(\mathbf{z}, d_t) = \{Y_{it}(z_i, d_t)\}_i.$$

The data that is available to the algorithm at time $t$ is a function of the data available at time $t-1$, as well as the observed outcomes at time $t-1$: $d_t(\mathbf{Y_{t-1}}(\mathbf{z}, d_{t-1}), d_{t-1}) = d_t(\mathbf{z}, d_{t-1})$. Since the treatment assignments $\mathbf{z}$ do not change over the course of the experiment, this suggests that the data available at any time $t$ can be expressed as $d_t^*(\mathbf{z}, d_1) := d_t(\mathbf{z}, d_{t-1}(\mathbf{z}, \ldots d_2(\mathbf{z}, d_1)))$. Hence, the observed outcome for unit $i$ at time $t$, $Y_{it} = Y_{it}(z_i, d_t^*(\mathbf{z}, d_1))$ is a function of unit $i$'s treatment, $Z_i$, as well as the training data available at time $t$, $d_t^*(\mathbf{z})$, which is itself a function of the full treatment assignment vector $\mathbf{z}$ and the data available at the beginning of the experiment $d_1$. In this notation, the total treatment effect at time $t$ of switching all recommendations from one algorithm to another for each unit $i$ can be expressed as

$$\tau_{it}^{TTE}(d_1) = Y_{it}(1, d_t^*(\mathbf{1}, d_1)) - Y_{it}(0, d_t^*(\mathbf{0}, d_1)).$$

Under naive experiment designs, neither quantity in this expression is observable, since for each unit of analysis we either observe $Y_{it}(1, d_t^*(\mathbf{z}, d_1))$ or $Y_{it}(0, d_t^*(\mathbf{z}, d_1))$ and in a randomized experiment $\mathbf{z}$ will not be $\mathbf{0}$ or $\mathbf{1}$. This presents a more severe obstacle to causal inference than the typical fundamental problem of causal inference.

In practical settings, and at the scale of global recommender systems, it is improbable for the action of each user to have a meaningful impact on the recommendations served to every other user. We postulate the existence of a network between units through which learning happens. It captures the intuition that the more similar two units are, the more the actions from one will inform the recommendations of the other. On the other hand, if two units are dissimilar, and are never or rarely liable to receive the same recommendations, their actions (and non-actions) will have negligible influence on

the other. The formalization of interference through a network is common in the relevant literature [4, 7, 12, 13, 18, 30, 47, 55, 57]. In our setting, this network is unobservable but can be approximated from the historical actions that units have taken, though we expect some bias in this approximation [10]. We briefly discuss how to conduct this approximation.

Unlike the majority of the interference literature, a recommender system setting may not have a directly observable graph between *units*; instead, the majority of interactions are between *users and items*, or users and creators. A subset of the literature [2, 7, 15, 21, 23, 31, 52, 64] focuses specifically on this bipartite graph setting, and assume the existence of a bipartite graph between units of two types (e.g. users and items), which is more easily observed (e.g. previous interactions of a user for a given recommended item). Brennan et al. [7] and Holtz et al. [23] propose approximating the unit-unit graph as a folding of the observed bipartite graph. The method of approximation of the underlying learning network is not the primary focus of this work, and the solutions discussed below do not assume that we know the graph exactly, as we do not expect unbiasedness. Instead, we show both theoretically and empirically that the bias under each design improves with properties of the unit-unit graph. In particular, the clustering solution can be made less biased than other baselines with only approximate knowledge of the unit-unit graph.

We now present an equilibrium network-based potential outcome model, which we find helpful to build intuition.[2] In particular, we no longer assume that outcomes are indexed by the data $d$ available to them; instead, the dependence is captured by the treatment assignment vector $\mathbf{z} = \{z_i\}$. Let $\mathbf{W} = \{w_{ij}\}_{i,j}$ denote the (possibly unobserved) weighted network through which learning happens, and $M$ denote the size of the corpus of items available to each recommender system, such that unit $i$'s outcome is given by

$$Y_i(\mathbf{z}) = \beta_{z_i} + \delta_{z_i} M + \sum_j w_{ij} \gamma_{z_i z_j} \tag{1}$$

where $\beta_z$ is a base treatment effect of algorithm $z$, $\delta_z$ is the positive effect of giving algorithm $z$ access to a larger corpus of content to recommend, and $\gamma_{z_i z_j}$ is an additional average effect of allocating a user to algorithm $z_j$ on the treatment effect of algorithm $z_i$ (a.k.a. an indirect effect).

The direct effect is captured by the coefficient $\beta_{z_i}$. The indirect effect of unit $j$ on $i$ is captured by $w_{ij} \gamma_{z_i z_j}$. The $\delta_{z_i} M$ parameter captures the linear relation observed in ablation studies between corpus size and outcomes [52, Fig. 5]. We could also consider a multiplicative model, whereby $Y_i(\mathbf{z}) = \delta_{z_i} M \left( \beta_{z_i} + \sum_j w_{ij} \gamma_{z_i z_j} \right)$. The formulation of the bias of the user-corpus co-diverted framework would change, but the conclusions would stay the same.

In practice, there may be many different (versions of) algorithms evaluated at any one time. We assume that the "treatment" ($z_i = 1$)

---

[2]Other potential outcome models have been considered in the literature [7, 8, 13, 18, 34]. For example, Brennan et al. [7] propose an alternative model directly on the bipartite user-item graph, where $Y_i = \alpha_i + \beta_i Z_i + \gamma_i \sum_j \sum_k v_{ik} v_{jk} Z_j$. In fact, $\sum_j \sum_k v_{ik} v_{jk}$ postulates a possible parameterization for the learning network $w_{ij}$, such that their model becomes $Y_i = \beta_{z_i} + \sum_j \gamma_i Z_j$, where $\beta_{z_i} := \alpha_i + \beta_i Z_i$. In this model, the indirect effect is unit-level heterogeneous, but does not depend on the treatment assignment. We find evidence for assignment-dependent indirect effects in our real data experiment in Section 5. We explore in Appendix A.2 an extension of their model to incorporate this additional heterogeneity. The takeaways do not change significantly.

algorithm is the one we are interested in evaluating, and all other algorithms are considered the "control" ($z_i = 0$), such that the total treatment effect is given by

$$\tau^{TTE} = (\beta_1 - \beta_0) + M(\delta_1 - \delta_0) + \frac{1}{N}\sum_i\sum_j w_{ij}(\gamma_{11} - \gamma_{00})$$

While many estimators are possible, we focus on the naive sample-mean estimator, which is often used by practitioners.[3]

$$\hat{\tau} = \frac{\sum_i Y_i 1\{Z_i = 1\}}{\sum_i 1\{Z_i = 1\}} - \frac{\sum_i Y_i 1\{Z_i = 0\}}{\sum_i 1\{Z_i = 0\}}$$

This is in line with previous work [7, 13, 29, 48] which also aims to study simple but possibly biased estimators through experimental designs, over more complex unbiased estimators. There are several reasons why one might want to adopt this strategy: firstly, unbiased estimators are more difficult to implement and often rely on additional assumptions about the network effects [18, 30, 37]; secondly, the sampling and design variances may be the dominant factor in the root mean-squared error of these unbiased estimators; finally, it is harder to "game" an analysis by emphasizing *design over analysis* [43]: design-first investigations lead to increased confidence in the results from non-experts, which has led to a strong tradition of using simple estimators and sophisticated designs in the tech industry. We now investigate the bias of this estimator under the different solutions proposed. Proofs and further discussions can be found in Appendix A.1.

*Independent assignment.* We first investigate the bias of a naive assignment that assigns units to treatment individually and identically. To ease the exposition, we will suppose the probability of treatment is exactly $\frac{1}{2}$ for each unit, but these results extend trivially to the general setting. The bias of the sample-mean estimator in this case is

$$Bias_{ind}(\hat{\tau}) \approx \frac{1}{N}\sum_i\sum_{j\neq i} w_{ij}\frac{1}{2}(-\gamma_{11} + \gamma_{10} - \gamma_{01} + \gamma_{00})$$

where the equality is given within $O(N^{-2})$ terms. In particular, If $\gamma_{11} = \gamma_{01} := \gamma_1$, $\gamma_{00} = \gamma_{10} := \gamma_0$, then this expression simplifies to $Bias_{ind}(\hat{\tau}) = \frac{1}{N}\sum_i\sum_{j\neq i} w_{ij}(\gamma_0 - \gamma_1)$. As expected, if algorithm 0 is better at generating useful information for learning, there is a positive bias in the estimated treatment effect, i.e. we over-estimate the effectiveness of algorithm 1.

*Bias of clustering.* If the treatment is assigned in a clustered way, with $C(.)$ being the cluster assignment function, this bias improves with a measure of clustering quality below, which is a variation of prior graph cut results [7, 13]:

$$Bias_{clu}(\hat{\tau}) \approx \frac{1}{N}\sum_i\sum_{j\neq i} w_{ij}\frac{1}{2}(-\gamma_{11} + \gamma_{10} - \gamma_{01} + \gamma_{00})1\{C(i) \neq C(j)\}$$

---

[3]The expectation of the sample-mean estimator under Bernoulli assignments is not immediately straightforward, due to the non-constant but highly concentrated normalizing constants $\sum_i 1\{Z_i = 1\}$ and $\sum_i 1\{Z_i = 0\}$. One option to express these results "cleanly" is to consider the non-individualistic completely randomized assignment, for which these are constant, or to consider the Horvitz-Thompson estimator [24], which normalizes each sum by $2/N$ here. In large samples, these considerations do not matter much, and lead to equal results within $O(N^{-2})$ terms. For ease of exposition, we ignore these terms here.

In particular, if $\gamma_{11} = \gamma_{01} := \gamma_1$, $\gamma_{00} = \gamma_{10} := \gamma_0$, the expression above reduces to:

$$Bias_{clu}(\hat{\tau}) \approx \frac{1}{N}\sum_i\sum_{j\neq i} w_{ij}(\gamma_0 - \gamma_1)1\{C(i) \neq C(j)\}$$

In other words, even if the learning effect is large ($|\gamma_1 - \gamma_0| \gg 0$), if cross-cluster dependence is small ($w_{ij} \approx 0$ for $C(i) \neq C(j)$), the bias will be small.

*Bias of data-diversion.* In a data-diverted experiment, each unit receives only a portion of the training data. For simplicity, we assume the cohort is split into two equal parts, which each receive half the training data. The resulting bias can also be easily expressed with our notation:

$$Bias_{div}(\hat{\tau}) \approx \frac{1}{N}\sum_i\sum_{j\neq i} w_{ij}\frac{\gamma_{00} - \gamma_{11}}{2}$$

The bias arises because each algorithm only learns from half of the sample. In some implementations [19, 52], each cohort shares a common core of data, to which is added a smaller share of exclusive traffic. In that case, the bias is an interpolation between the bias under the independent assignment and the formula above.

*Bias of user-corpus co-diversion.* In a user-corpus co-diverted experiment, both users and items are randomized to treatment or control, and treated (control) users only have access to treated (control) items. Again for simplicity, we assume that both users and items are split into two equal parts. The resulting bias can be expressed with our notation as:

$$Bias_{codiv}(\hat{\tau}) \approx \frac{1}{N}\left(\sum_i\sum_{j\neq i} w_{ij}\frac{\gamma_{00} - \gamma_{11}}{2}\right) + M\left(\frac{\delta_0 - \delta_1}{2}\right)$$

The bias arises both because each algorithm only learns from half as much data, *and* because each algorithm only has access to half as many items.

*Comparison of bias.* The design leading to the smallest amount of bias depends on the quality of clusters, the strength of learning effects, and the extent to which each algorithm's performance scales with the size of the item corpus. For the sake of expositional simplicity, we compare the bias of clustering and data-diversion, but both designs can also easily be compared to user-corpus co-diversion. The difference between the bias of the data-diverted and clustered treatment effect estimates is:

$$Bias_{div}(\hat{\tau}) - Bias_{clu}(\hat{\tau}) \approx$$
$$\frac{1}{2N}\sum_i\sum_{j\neq i} w_{ij}(-\gamma_{11} + \gamma_{00})1\{C(i) = C(j)\}$$
$$- \frac{1}{2N}\sum_i\sum_{j\neq i} w_{ij}(\gamma_{10} - \gamma_{01})1\{C(i) \neq C(j)\}$$

Since the sign of the biases cannot be determined, it is difficult to ascertain which bias is smaller. If $\gamma_{11} = \gamma_{01} := \gamma_1$, $\gamma_{00} = \gamma_{10} := \gamma_0$, i.e. the information generated by one algorithm benefits both algorithms equally, then $sign(Bias_{div}(\hat{\tau})) = sign(Bias_{clu}(\hat{\tau}))$. Under this assumption, without loss of generality, we can assume

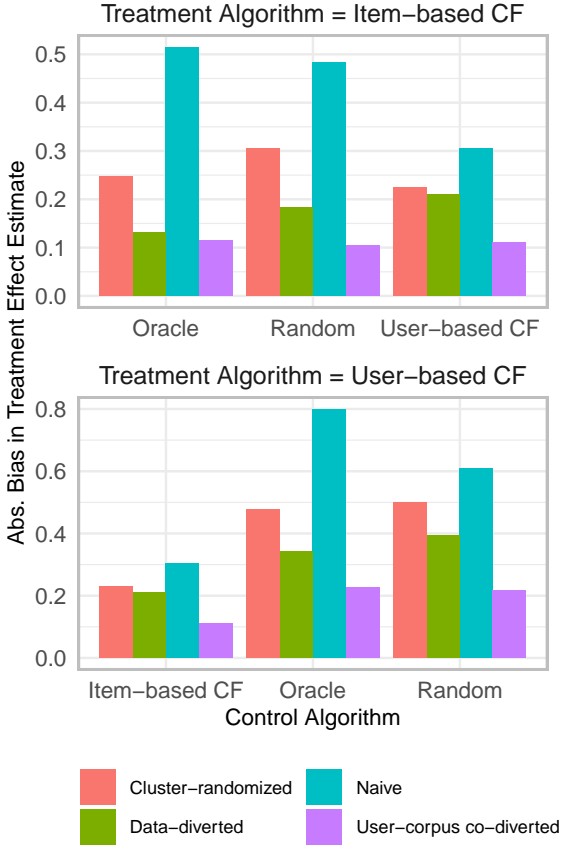

**Figure 2: The estimated absolute bias of treatment effect estimates for different algorithm pairs across various experimental designs. The results use $\gamma_{pref} = 10$, with all other simulation parameters set to their default values.**

$\gamma_0 - \gamma_1 = 1$. In that case, the bias simplifies further,

$$Bias_{div}(\hat{\tau}) - Bias_{clu}(\hat{\tau}) \approx \frac{1}{2N} \sum_i \sum_{j \neq i} w_{ij} 1\{C(i) = C(j)\}$$

$$- \frac{1}{2N} \sum_i \sum_{j \neq i} w_{ij} 1\{C(i) \neq C(j)\}$$

In other words, the information generated by one algorithm benefits both algorithms equally, *and* if the clusters are of high-quality (low cross-cluster dependence), then the clustering experiment performs better than the data-diverted experiment. Another possible setting is that the information generated and used by the two algorithms is orthogonal, i.e. $\gamma_{10} = \gamma_{01} = 0$. In that case,

$$Bias_{div}(\hat{\tau}) - Bias_{clu}(\hat{\tau}) \approx \frac{1}{2N} \sum_i \sum_{j \neq i} w_{ij} (\gamma_{00} - \gamma_{11}) 1\{C(i) = C(j)\}$$

Here, if $\gamma_{00} > \gamma_{11}$, then the clustering algorithm is always preferred; otherwise, the data-diverted solution is preferred.

In addition to bias, an important consideration when choosing amongst these experimental designs is the variance of the resulting treatment effect estimate. We provide a brief discussion of this matter in Appendix B.

## 4 SIMULATING SYMBIOSIS BIAS

Although the theoretical model analyzed in this paper provides useful insight into the efficacy of different experiment designs at reducing symbiosis bias, one might worry that the model abstracts away the complex inter-temporal dynamics that cause symbiosis bias. To address this concern, we use a simple simulation framework to document the existence of symbiosis bias in naive experiments, characterize the efficacy of different experiment designs at reducing symbiosis bias, and conduct a preliminary exploration into the conditions under which symbiosis bias may be more or less severe.

What follows is a high-level description of our simulation framework that captures its important elements. A more comprehensive description of this framework, which is inspired by the simulation framework presented in Chaney et al. [9], is offered in Appendix C. We consider a community of 100 users interacting with 1,000 items over the course of 100 time periods. All 1,000 items are not initially available to users; instead, items are made available in increments of 10 items per time period, so that there are 10 items available at $t = 1$ and 1,000 items available at $t = 100$. The staggered release of items ensures that there are always new items about which a data-based recommender system will have limited information. User preferences and item attributes are represented by 10-dimensional vectors $\rho_u$ and $v_i$, respectively, that are both drawn from nearly-symmetric Dirichlet distributions that are distorted so as to create some natural clustering in both the preferences of users and the attributes of items. User $u$'s actual utility from consuming item $i$ is $Beta(\mu = \rho_u^T \alpha_i, \sigma = 10^{-5})$. However, user $u$ places a premium on consuming objects that are more highly recommended by the platform's recommendation algorithm. At each time interval $t$, all items available to a user $u$ are ranked and presented to them. The user either interacts the item that they perceive will bring them the highest utility, or they interact with no item at all if no item in the consideration set provides higher than the median utility offered across all items (including those not in the present consideration set). Each user can consume each item $i$ at most once. Each user's potential outcome is the rate at which he or she chooses to interact with items, taken over the $T = 100$ time periods.

After the first $t_{init} = 10$ periods of the simulation, during which all recommendations are random, the user population is randomized into an experiment that compares two different recommendation algorithms. We consider four different experiment designs, all of which have been discussed previously in this paper: naive experimentation, cluster-randomized experimentation, data-diverted experimentation, and user-corpus co-diverted experimentation. Using these designs, we consider pairwise combinations of four possible recommendation algorithms:

- **Oracle**: An algorithm that recommends the available content to each user that will give them the highest utility.
- **Random**: An algorithm that randomly recommends content to users.

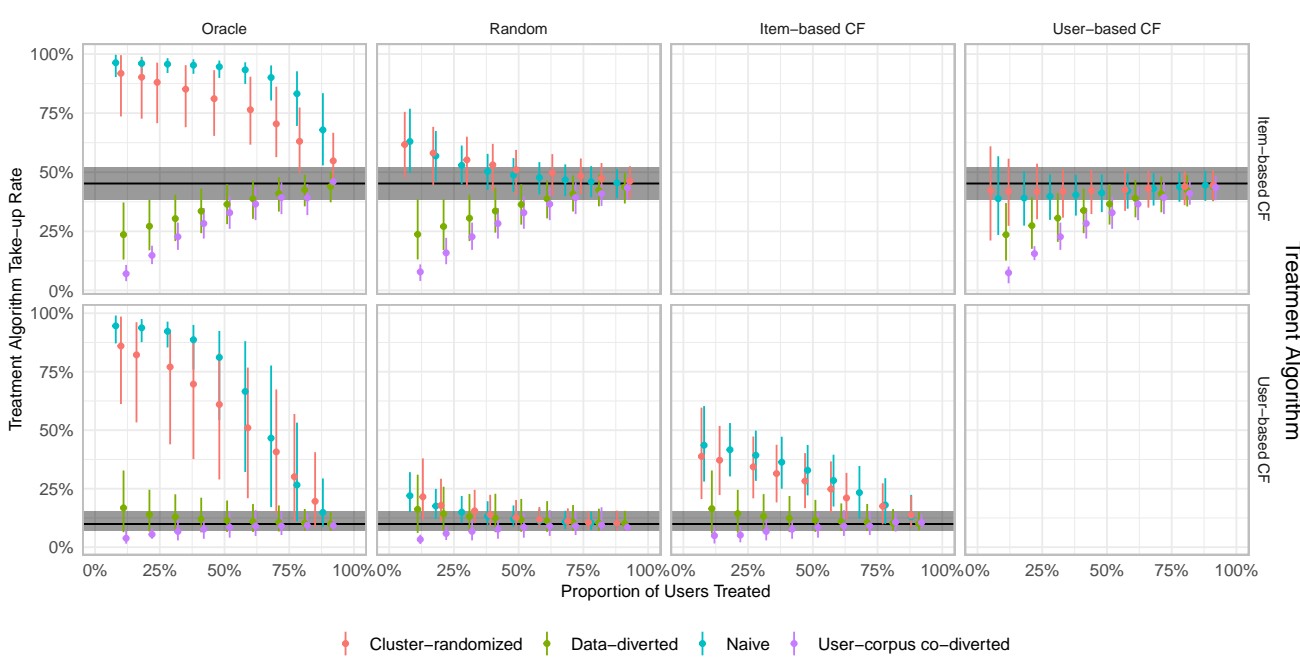

**Figure 3: The estimated take-up rate for different treatment algorithms when tested against different control algorithms under different experiment designs at different levels of treatment-control. Error bars indicate 95% confidence intervals. The black line indicates the "true" take-up rate for each control algorithm, which the shaded area representing the 95% confidence interval.**

- **Item-based collaborative filter**: An algorithm that recommends items most similar to the items that a user has previously consumed.
- **User-based collaborative filter**: An algorithm that recommends items consumed by the users most similar to the focal user.

We also use our simulation framework to measure the "true" counterfactual take-up rate under each of these recommendation algorithms.[4]

Figure 2 shows the estimated absolute bias in the treatment effect estimate obtained when comparing different pairs of algorithms under different experiment designs. The results use $\gamma_{pref} = 10$, with all other simulation parameters set to the default values described in Appendix C. All three experiment designs reduce the absolute bias of treatment effect estimates relative to the naive experiment design, with the relative efficacy of the different designs varying depending on the particular pair of algorithms being compared. However, at least for this particular value of $\gamma_{pref}$, it appears to be the case that user-corpus co-diversion is the most effective design for reducing symbiosis bias, whereas clustering is the least. The relative efficacy of each experiment design also varies as a function of $\gamma_{pref}$; this is discussed in more detail in Appendix D.

We can also use our framework to decompose the TTE bias into bias in individual treatment arms' take-up rates. Figure 3 shows

the estimated take-up rate for each algorithm when compared to different competitor algorithms, under different experiment designs, under different levels of treatment-control split, along with the true counterfactual take-up rate for each algorithm. A number of interesting insights emerge. Firstly, in the majority of cases, naive experimentation does lead to symbiosis bias, and in the most severe cases, the magnitude of this bias can be large (e.g., exaggerating the take-up rate by nearly an order of magnitude). Second, the fact that the magnitude of this bias decreases as the percentage of units treated increases indicates that in many (but not all) cases, symbiosis bias arises because one algorithm benefits from the "free" exploration provided by the other algorithm. Third, while clustering is not as effective as data-diversion and user-corpus co-diversion at reducing absolute bias, it does not introduce new, opposite-signed bias arising from less data and/or a smaller item corpus. Finally, it is worth noting that clustering does indeed come at the expense of lower precision, as evidenced by the wider confidence intervals around clustered take-up rate estimates.

## 5 REAL DATA APPLICATION: A COUNTRY-DIVERTED EXPLORATION EXPERIMENT

While prior work has used data to empirically evaluate the efficacy of data-diverted and user-corpus co-diverted experiments [11, 52], cluster randomization has as of yet not been evaluated as a method for symbiosis bias reduction. In this section, we demonstrate the

---

[4]This is achieved by conducting a naive experiment in which the treatment and control algorithms are the same.

presence of symbiosis bias with evidence from a large industrial recommender system, and use this same data to gain insight into the extent to which cluster randomization can mitigate symbiosis bias. We study a 25% country-diverted A/B test described by Lin et al. [35], in which the treatment increased exploration by boosting recommendations of recently published content to treated users. Countries act as natural clusters to the extent that users engage with similar content as others in their country. These clusters are imperfect, since users in different countries do engage with some of the same content.

Our hypothesis for the mechanism of symbiosis bias in this experiment is as follows: treated users are recommended more recently published content. This results in more training data on this content in the shared training data pool, as the algorithm learns which recently published content is appealing to users. Users in the control condition who have similar interests to those in the treated condition are recommended recently published content because of the training data generated by the treated condition. This leads to an increase in the consumption of recently published content for control users, with a greater increase for control users whose interests overlap strongly with treated users.

To test this hypothesis, we study the correlation between the amount of recently published content each country views and its aggregate exposure to treated countries. This data is shown in Figure 4. Our definition of exposure uses the co-engagement metric introduced in [7], which approximates the unit-unit graph as described in the footnote of Section 3. We observe that engagement with recently published content correlates positively with a control country's exposure to treatment. This correlation supports the hypothesis that exploration data from treated countries "leaks" into the training data for similar control countries, causing the recommender system to recommend more recently published content to users in these countries. This correlation also highlights to potential for cluster randomization to reduce symbiosis bias, and the extent to which cluster randomization's efficacy as a bias reduction technique is dependent on cluster quality. We also observe that the correlation is near zero for treated countries. This negligible impact of exposure on treated countries further illustrates the asymmetric nature of symbiosis bias.

Overall, these results align with our potential outcomes model from Equation 1, supporting both the linear effect of exposure on the outcome and the interaction between exposure and treatment status.

## 6 DISCUSSION

In this paper, we have used theory and simulation to explore the efficacy of three different experiment designs at reducing *symbiosis bias*, a novel form of bias in A/B test treatment effect estimates that arises when two algorithms share a common pool of training data. Our results reveal that each of the three approaches considered (cluster randomization, data-diversion, and user-corpus co-diversion) have benefits and drawbacks. The extent to which symbiosis bias impacts a given experiment, and the relative efficacy of different experiment designs at reducing this bias, depends on a number of factors, including the specific algorithms being compared, the percentage of site traffic available for enrollment in an experiment,

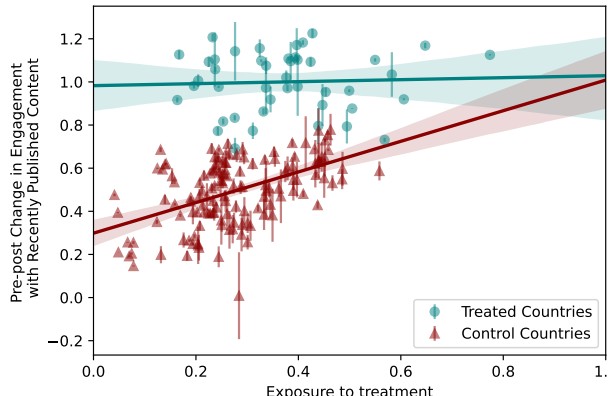

Figure 4: Country-level differences in pre-period vs experimental outcomes are linear in exposure for control and for treated countries, matching our potential outcomes model of symbiosis bias. Small countries omitted for readability. Error bars represent 95% confidence intervals. Y-axis is normalized so the average treated outcome equals 1.

the quality of clusters that can be inferred using available data, and the size of the corpus of items being recommended.

Our work has a number of important limitations. For instance, our theoretical model does not account for heterogeneity. Furthermore, our simulation framework and the analyses conducted using this framework can be expanded in numerous ways (i.e., looking at a larger set of recommendation algorithms and simulation parameters). Finally, our analysis of real data relies on pre-post analysis of a country-diverted experiment, and does not provide fully rigorous well-identified evidence of the relationship between cluster quality and symbiosis bias reduction. Nonetheless, we believe our analyses provide useful insights on a research topic that is still nascent.

We also see numerous promising areas for future work, including the development of more intelligent methods for splitting data and items under data-diverted and user-corpus co-diverted designs, making further comparisons between the approaches considered in this paper and other, analysis-based approaches to reducing symbiosis bias (e.g., Goli et al. [17] and Si [49]), and/or conducting meta-experiments in the style of Holtz et al. [23] or Saveski et al. [45].

In conclusion, this paper has provided a detailed investigation of symbiosis bias in A/B testing of recommendation algorithms, examining three experimental designs—cluster randomization, data-diversion, and user-corpus co-diversion. Through theoretical models, simulations, and real-world data, we have demonstrated that each approach has its own strengths and limitations, depending on the context and algorithms being tested. While our findings offer practical insights for mitigating symbiosis bias, future research is needed to further refine these methods and explore other bias-reduction strategies for A/B tests that compare recommendation algorithms.

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

# A THEORY RESULTS

## A.1 Proofs of Results in Section 3

For all results, we ignore the $O(N^{-2})$ terms that result from the $\sum_i 1\{Z_i = 1\}$ and $\sum_i 1\{Z_i = 0\}$ being non-constant. Recall the potential outcome model:

$$Y_i(\mathbf{z}) = \beta_{z_i} + \delta_{z_i}M + \sum_j w_{ij}\gamma_{z_i z_j}$$

The average total treatment effect is given by:

$$\begin{aligned}
\tau^{TTE} &= \mathbb{E}[Y_{i2}|\mathbf{Z} = \mathbf{1}] - \mathbb{E}[Y_{i2}|\mathbf{Z} = \mathbf{0}] \\
&= (\beta_1 - \beta_0) + M(\delta_1 - \delta_0) + \frac{1}{N}\sum_i\sum_j w_{ij}(\gamma_{11} - \gamma_{00})
\end{aligned}$$

The expression of the sample-mean estimator is given by:

$$\hat{\tau} = \frac{\sum_i Y_{i2}1\{Z_i = 1\}}{\sum_i 1\{Z_i = 1\}} - \frac{\sum_i Y_{i2}1\{Z_i = 0\}}{\sum_i 1\{Z_i = 0\}}$$

For an independent assignment with prob $\frac{1}{2}$,

$$\begin{aligned}
\mathbb{E}_{ind}[\hat{\tau}] =&(\beta_1 - \beta_0) + M(\delta_1 - \delta_0) \\
&+ \frac{1}{N}\sum_i w_{ii}(\gamma_{11} - \gamma_{00}) \\
&+ \frac{1}{N}\sum_i\sum_{j\neq i} w_{ij}\frac{1}{2}(\gamma_{11} + \gamma_{10} - \gamma_{01} - \gamma_{00})
\end{aligned}$$

The bias then becomes:

$$\begin{aligned}
Bias_{ind}(\hat{\tau}) &\equiv \mathbb{E}_{ind}[\hat{\tau}] - \tau_2^{TTE} \\
&= \frac{1}{N}\sum_i\sum_{j\neq i} w_{ij}\frac{1}{2}(-\gamma_{11} + \gamma_{10} - \gamma_{01} + \gamma_{00})
\end{aligned}$$

We now consider a clustering experiment with $C(.)$ being the cluster assignment function. We have:

$$\begin{aligned}
\mathbb{E}_{clu}[\hat{\tau}] =& (\beta_1 - \beta_0) + M(\delta_1 - \delta_0) \\
&+ \frac{1}{N}\sum_i\sum_j w_{ij}(\gamma_{11} - \gamma_{00})1\{C(i) = C(j)\} \\
&+ \frac{1}{N}\sum_i\sum_j w_{ij}\frac{1}{2}(\gamma_{11} + \gamma_{10} - \gamma_{00} - \gamma_{01})1\{C(i) \neq C(j)\}
\end{aligned}$$

The bias is then given by:

$$\begin{aligned}
Bias_{clu}(\hat{\tau}) &\equiv \mathbb{E}_{clu}[\hat{\tau}] - \tau_2^{TTE} \\
&= \frac{1}{N}\sum_i\sum_{j\neq i} w_{ij}\frac{1}{2}(-\gamma_{11} + \gamma_{10} - \gamma_{01} + \gamma_{00})1\{C(i) \neq C(j)\}
\end{aligned}$$

We now consider a data-diverted experiment:

$$\begin{aligned}
\mathbb{E}_{div}[\hat{\tau}] =& (\beta_1 - \beta_0) + M(\delta_1 - \delta_0) \\
&+ \frac{1}{N}\sum_i w_{ii}(\gamma_{11} - \gamma_{00}) \\
&+ \frac{1}{N}\sum_i\sum_{j\neq i} w_{ij}\frac{1}{2}(\gamma_{11} - \gamma_{00})
\end{aligned}$$

The bias is then given by:

$$Bias_{div}(\hat{\tau}) \equiv \mathbb{E}_{div}[\hat{\tau}] - \tau_2^{TTE} = \frac{1}{N}\sum_i\sum_{j\neq i} w_{ij}\frac{1}{2}(-\gamma_{11} + \gamma_{00})$$

Finally, we consider a user-corpus co-diverted framework. The expectation is given by:

$$\begin{aligned}
\mathbb{E}_{codiv}[\hat{\tau}] =&(\beta_1 - \beta_0) + \frac{1}{N}\sum_i w_{ii}(\gamma_{11} - \gamma_{00})+ \\
&\frac{1}{N}\sum_i\sum_{j\neq i} w_{ij}\frac{1}{2}(\gamma_{11} - \gamma_{00}) + M\left(\frac{\delta_1 - \delta_0}{2}\right)
\end{aligned}$$

As a result, the bias is given by:

$$Bias_{codiv}(\hat{\tau}) \approx \frac{1}{N}\left(\sum_i\sum_{j\neq i} w_{ij}\frac{\gamma_{00} - \gamma_{11}}{2}\right) + M\left(\frac{\delta_0 - \delta_1}{2}\right)$$

## A.2 Alternative Potential Outcome Model

[7] propose the following potential outcome model in a bipartite graph:

$$[\mathbf{B}] \quad Y_i = \alpha_i + \beta_i Z_i + \gamma_i \sum_j\sum_k v_{ik}v_{jk}Z_j$$

[B] is not a suitable model in our real data setting: whether a unit is treated or controlled affects how much interference they receive, e.g. a random algorithm does not learn from other algorithms' findings. We can extend this model by adding an additional $\delta_i Z_i$ term (the corpus-dependence term is irrelevant for the computations below).

$$[\mathbf{C}] \quad Y_i = \alpha_i + \beta_i Z_i + (\gamma_i + \delta_i Z_i)\sum_j\sum_k v_{ik}v_{jk}Z_j$$

We answer two questions: (1) Is this parameterization as expressive as the model in Eq. 1? (2) What is the bias-minimizing clustering (given fixed cluster cardinalities)? To answer the first question, we rewrite our model (1) as a polynomial in $Z_i$ and $Z_j$:

$$\begin{aligned}
Y_i =& \underbrace{\alpha_i' + \delta_{00}\sum_j w_{ij}}_{\alpha_i} + \underbrace{\left[\beta_i' + (\delta_{10} - \delta_{00})\sum_j w_{ij}\right]}_{\beta_i} Z_i \\
&+ \sum_j Z_j \underbrace{w_{ij}(\delta_{01} - \delta_{00})}_{\gamma_i\sum_k v_{ik}v_{jk}} + Z_i\sum_j Z_j\underbrace{w_{ij}(\delta_{11} - \delta_{10} - \delta_{01} + \delta_{00})}_{\delta_i\sum_k v_{ik}v_{jk}}
\end{aligned}$$

We can solve for $\delta_{00}, \delta_{01}, \delta_{10}, \delta_{11}$:

$$\delta_{00} = (\alpha_i - \alpha_i')\left(\sum_j w_{ij}\right)^{-1}$$

$$\delta_{10} = (\beta_i - \beta_i' + \alpha_i - \alpha_i')\left(\sum_j w_{ij}\right)^{-1}$$

$$\delta_{01} = \gamma_i w_{ij}^{-1}\sum_k v_{ik}v_{jk} + \delta_{00}, \quad \forall j$$

$$\begin{aligned}
\delta_{11} &= \delta_{10} + \delta_{01} - \delta_{00} + \delta_i w_{ij}^{-1}\sum_k v_{ik}v_{jk} \\
&= \delta_{10} + (\delta_i + \gamma_i)w_{ij}^{-1}\sum_k v_{ik}v_{jk}
\end{aligned}$$

If we maintain that $\delta_{00}, \delta_{01}, \delta_{10}, \delta_{11}$ are constants with respect to $j$, then it follows from the $\delta_{01}$ or the $\delta_{11}$ equations that $w_{ij}^{-1}\sum_k v_{ik}v_{jk}$ is constant with respect to $j$, such that

$$\exists c_i, \forall j, \ w_{ij} = c_i \sum_k v_{ik} v_{jk}.$$

In that case, the above equations for $\delta_{10}, \delta_{11}$ simplify to:

$$\delta_{01} = \gamma_i c_i^{-1} + \delta_{00}$$
$$\delta_{11} = \delta_{10} + (\delta_i + \gamma_i)\, c_i^{-1}$$

If we also maintain that $\delta_{00}, \delta_{01}, \delta_{10}, \delta_{11}$ are constant with respect to $i$ and $j$, then:

$$\exists C_0, \forall i, \alpha_i - \alpha_i' = C_0 \sum_j w_{ij} \quad \text{(from } \delta_{00})$$
$$\exists C_1, \forall i, \beta_i - \beta_i' = C_1 \sum_j w_{ij} \quad \text{(from } \delta_{00} \text{ and } \delta_{10})$$
$$\exists C_3, \forall i, \gamma_i = C_3 c_i \quad \text{(from } \delta_{01})$$
$$\exists C_4, \forall i, \delta_i = C_4 c_i \quad \text{(from } \delta_{01} \text{ and } \delta_{11})$$

This means that our model (1) can be re-written in the following way:

$$Y_i = \alpha_i + \beta_i Z_i + (C_3 + C_4 Z_i) \sum_j w_{ij} Z_j$$

In particular, this means that the ratio of the indirect effect on a treated unit and the indirect effect on a controlled unit is constant across all units and equal to $1 + \frac{C_4}{C_3}$, which is a strong assumption. This assumption is clearly present in the original [YT] model. To answer question 1, [YTB1] is more expressive than [YT].

To answer the second question, we take the expectation of the model under a cluster-randomized experiment with fixed cardinality, and isolate its bias relative to the Total Treatment Effect. To simplify the notation, we will denote: $w_{ij} = \sum_k v_{ik} v_{jk}$. Recall that the total treatment effect of [YTB1] is given by:

$$TTE = \beta_i + (\gamma_i + \delta_i) \sum_j w_{ij}$$

Let $C_i$ denote unit $i$'s cluster.

$$\mathbb{E}\left[Y_i Z_i p^{-1} \middle| C\right] = \alpha_i + \beta_i + \sum_j w_{ij}(\gamma_i + \delta_i)\mathbb{P}(Z_i Z_j = 1)p^{-1}$$

$$\mathbb{E}[Y_i(1 - Z_i)(1 - p)^{-1} | C] = \alpha_i + \sum_j w_{ij}\gamma_i \mathbb{P}((1 - Z_i)Z_j = 1)(1 - p)^{-1}$$

Furthermore, let $\lfloor pK \rfloor$ be the number of treated clusters. For simplicity, we will assume $\lfloor pK \rfloor = pK$. We have:

$$\mathbb{P}(Z_i Z_j = 1)p^{-1} = 1_{C_i = C_j} + 1_{C_i \neq C_j} p(1 - K^{-1})$$
$$\mathbb{P}((1 - Z_i)Z_j = 1)(1 - p)^{-1} = p 1_{C_i \neq C_j}$$

Assuming $K \gg 1$, the expectation and bias of the diff-in-means estimator is:

$$\mathbb{E}[\hat{\tau}] = \sum_i \left( \beta_i + \sum_j w_{ij} \left[ \gamma_i 1_{C_i = C_j} + \delta_i \left(1_{C_i = C_j} + 1_{C_i \neq C_j} p\right) \right] \right)$$

$$TTE - \mathbb{E}[\hat{\tau}] = \sum_i \sum_j w_{ij} \left( \gamma_i 1_{C_i \neq C_j} + \delta_i(1 - p)1_{C_i \neq C_j} \right)$$

$$= \sum_i \sum_j w_{ij} \left( \gamma_i + \delta_i(1 - p) \right) 1_{C_i \neq C_j}$$

The answer to the second question is that it is again a graph cut, where the edge weights are reweighted by $\gamma_i + \delta_i(1 - p)$. As a result, the conclusions around the effectiveness of clustering do not change much with this alternative model. The same holds for other designs.

## B  VARIANCE OF TREATMENT EFFECT ESTIMATORS

An important consideration when choosing amongst these experimental designs is the variance of the chosen estimator. It is surprisingly non-straightforward and verbose to compute each variance under linear interference models [3, 8, 34]. We provide some intuition on the variance of each mechanism where the indirect effects are second-order to the first-order effects, such that we can reasonably assume that the variance behaves as though SUTVA holds. The Bernoulli assignment, the user-corpus co-diverted experiment, and the data diversion mechanism all consider an equal number of unit-level outcomes. As a result, we expect each of these three designs to have roughly similar RMSE. The cluster-randomized assignment can be framed as a Bernoulli assignment where individual outcomes are replaced with cluster-level outcomes[5]. The effective number of units goes from $N$ to $C$, the number of clusters. Under SUTVA, the standard deviation of our estimate under a Bernoulli assignment decreases roughly at a rate of $N^{-1/2}$, such that, if the indirect effects are second-order to the direct effects, we expect the RMSE to grow roughly at a rate of $N^{1/2}C^{-1/2}$ when going from a Bernoulli assignment with $N$ to a cluster-randomized assignment with $C$ balanced clusters. The standard deviation is expected to grow even further with unbalanced clusters. In practice, the variance of our estimators is often dominated by the sampling variance of outcomes, and not by the treatment effects, such that practitioners can use the variance of each estimator in an A/A test as a reasonable estimate of each variance.

## C  SIMULATION FRAMEWORK DESCRIPTION

In the following subsections, we provide a detailed description of each component of our simulation framework. In our paper's simulations, the default values of the simulation parameters are as follows: $p = 0.5$, $N_{C_i} = 4$, $N_{C_u} = 10$, $\alpha_u = 1$, $\alpha_i = 0.01$, $\gamma_{item} = 1$, $\gamma_{pref} = 1$, $T = 100$, $t_{init} = 10$, $d = 0.8$, $f = 1$, $n_{items} = 1,000$, and $n_{users} = 100$.

### C.1  User Preferences

We represent user preferences as 10-dimensional vectors, $\rho_u$, whose entries sum to 1. These vectors are drawn from a modified symmetric Dirichlet distribution, where one out of $N_{C_u}$ possible components is scaled by a random parameter, $\gamma_{pref}$. Specifically, we begin with a Dirichlet distribution for vectors of length 10 with concentration parameters $(\alpha_1, \alpha_2, \ldots, \alpha_{10})$. Initially, the distribution is symmetric with all concentration parameters equal, i.e., $(\alpha_1 = \alpha_2 = \cdots = \alpha_{10} = \alpha_u)$. We then select $N_{C_u}$ of the concentration parameters, which correspond to the "clusters" in our

---

[5]In other words, $Y_i \leftarrow Y_c^+ := \beta_{z_c}^+ + \sum_{c'} w_{cc'} \gamma_{z_c} z_{c'}$, where $\beta_{z_i} \leftarrow \beta_{z_c}^+ := |\{i \in C\}| \cdot \beta_{z_c}$ and $w_{ij} \leftarrow w_{cc'} := \sum_{i \in c, j \in c'} w_{ij}$.

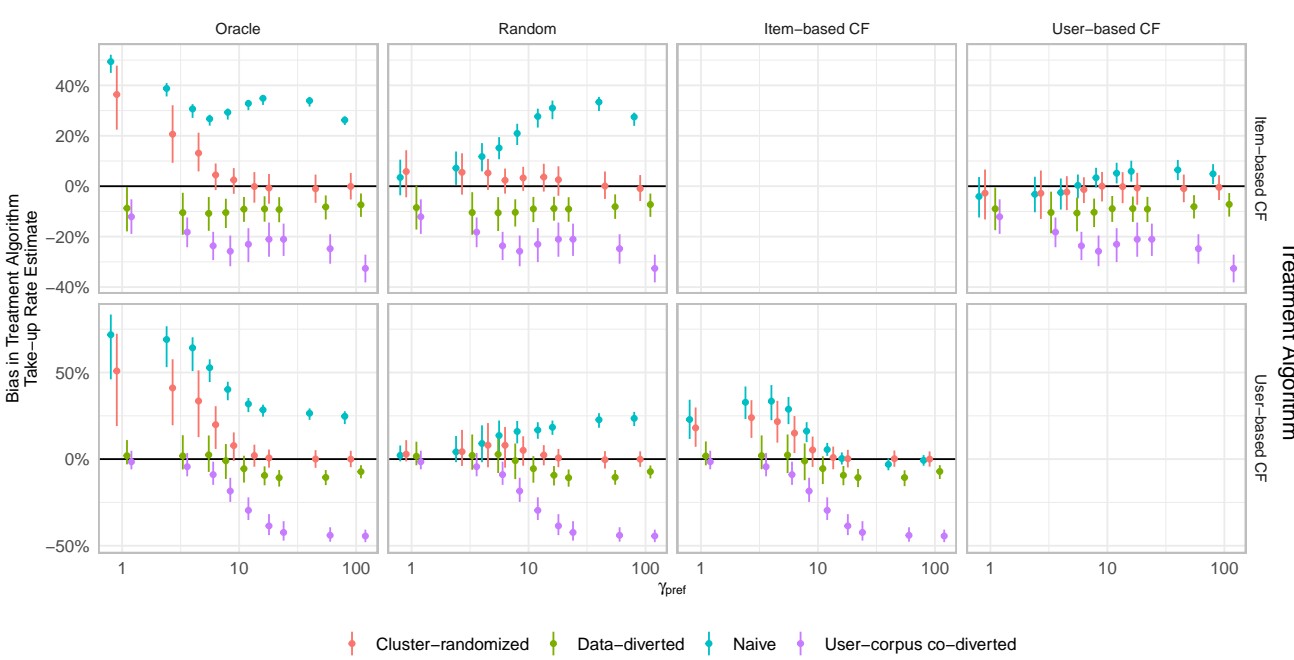

**Figure 5: Bias in the estimated take-up rate (relative to the mean "true" take-up rate) for different treatment algorithms when tested against different control algorithms under different experiment designs and under different values of $\gamma_{pref}$ (which determines how easily clustered user preferences are). Error bars indicate 95% confidence intervals.**

data-generating process (for instance, if $N_{C_i}$ were 3, the three selected concentration parameters might be $\alpha_1$, $\alpha_7$, and $\alpha_{10}$). Each user randomly draws one of the $N_{C_u}$ "clusters", and then draws their preferences from a modified Dirichlet distribution where the selected concentration parameter is equal to $\gamma_{pref} \cdot \alpha_u$, rather than $\alpha_u$.

## C.2 Item Attributes

We represent item attributes as 10-dimensional vectors, $v_i$, whose entries sum to 1. These vectors are drawn from a modified symmetric Dirichlet distribution, where one out of $N_{C_u}$ possible components is scaled by a random parameter, $\gamma_{item}$. Specifically, we begin with a Dirichlet distribution for vectors of length 10 with concentration parameters $(\alpha_1, \alpha_2, \ldots, \alpha_{10})$. Initially, the distribution is symmetric with all concentration parameters equal, i.e., $(\alpha_1 = \alpha_2 = \cdots = \alpha_{10} = \alpha_i)$. We then select $N_{C_i}$ of the concentration parameters, which correspond to the "clusters" in our data-generating process (for instance, if $N_{C_u}$ were 3, the three selected concentration parameters might be $\alpha_1$, $\alpha_7$, and $\alpha_{10}$). Each user randomly draws one of the $N_{C_u}$ "clusters", and then draws their preferences from a modified Dirichlet distribution where the selected concentration parameter is equal to $\gamma_{item} \cdot \alpha_i$, rather than $\alpha_i$.

## C.3 User Consumption Decisions

User $u$'s true utility from consuming item $i$ is drawn from

$$util(\rho_u, v_i) \sim Beta(\mu = \rho_u^T v_i, \sigma = 10^{-5})$$

However, user $u$ does not know how much utility they will derive from consuming item $i$ prior to consumption. User $u$'s *perceived* utility from consuming item $i$, $util_p(\rho_u, v_i)$, depends on item $i$'s ranking $r$ in the set of results presented to user $u$, with $r = 1$ corresponding to the most highly ranking item. This perceived utility is equal to:

$$util_p(\rho_u, v_i) = util(\rho_u, v_i) \cdot r^d$$

In other words, users expect more highly ranked items to be of higher quality.

## C.4 Simulated Experiment Design

In each simulation, we begin with $n_{items}$ items and $n_{users}$ users, with item attributes and user preferences generated as described in the previous sections. Each user also has a reserve utility, $reserve_u$, which is equal to the median of the true utilities associated user $u$ consuming each of all $n_i tems$ items.

The simulation lasts $T$ time periods. At the beginning of the simulation, no items are available for consumption by users. In each period, beginning with period $t = 1$, $\frac{n_{items}}{t}$ randomly selected items are made available for user consumption, and are available to

be consumed in all subsequent periods. Each user $u$ can consume each item $i$ a maximum of one time.

For the first $t_{init}$ time periods, all available items are suggested to users according to a random ranking. After $t_{init}$ time periods, users (and items in the case of the user-item co-diverted experiment) are randomized into two different ranking algorithms according to one of four experiment designs: naive experimentation, clustered experimentation, data-diverted experimentation, or user-corpus co-diverted experimentation, with $p \times n_{users}$ being allocated to the treatment and $(1-p) \times n_{users}$ being allocated to the control.[6] In each subsequent time period, all previously available items are ranked according to the user's assigned ranking algorithm. The $\frac{n_{items}}{t}$ new items, for which there is no historical data, are then randomly ordered and interleaved into the ranking algorithm's ordered list. Each ranking algorithm is re-trained using the newest available every $f$ time periods.

At the end of each simulated experiment, we calculate the average rate at which users consumed items in each treatment arm, as well as the difference between these consumption rates, i.e., the treatment effect.

## D  BIAS IN TAKE-UP RATE ESTIMATES AS A FUNCTION OF $\gamma_{pref}$

Figure 5 shows how the amount of symbiosis bias changes under each experiment design as a function of $\gamma_{pref}$, which is a parameter in our simulation that dictates how easily clustered user preferences are. Higher values of $\gamma_{pref}$ correspond to more separated user preferences and better clusters. The amount of symbiosis bias in estimates of each algorithm's take-up rate is highly dependent on not only the combination of algorithms being compared, but also the value of $\gamma_{pref}$. For instance, for low-values of $\gamma_{pref}$, naive and clustered randomization lead to the highest bias in estimates of the user-based CF take-up rate when compared to the item-based CF, whereas for high values of $\gamma_{pref}$, the user-corpus co-diverted experiment is the most biased.

---

[6]In the case of cluster-randomized experimentation, these user-level treatment assignment probabilities are approximate.