# OpenReview forum: "Reducing Symbiosis Bias Through Better A/B Tests of Recommendation Algorithms"
_ACM.org/TheWebConf/2025/Conference — WWW 2025 Poster_

### Official Review · Reviewer_ojuP · 2024-11-25

**Novelty:** 3
**Technical Quality:** 2

**Review:**

This paper focuses on the issue of "symbiotic bias" in A/B testing for recommendation systems and proposes three experimental designs to address this challenge. Overall, the paper clearly identifies the sources of bias and demonstrates the effectiveness of the proposed methods through theoretical modeling, simulation experiments, and real-world data validation. The study introduces cluster randomized design into the context of symbiotic bias for the first time, contributing a novel experimental design approach to the field. Furthermore, by combining theoretical insights with empirical studies, the paper offers practical guidance for experimental design in recommendation algorithms. However, the level of innovation in the work is not entirely clear, and both the models and experiments would benefit from further refinement.

Some assumptions in the theoretical model may be overly simplified (e.g., "Again for simplicity, we assume that both users and items are split into two equal parts"), which could limit the generalizability of the model.

The paper lacks direct empirical comparison experiments with existing analytical methods, leaving the relative advantages and disadvantages of cluster randomized design in practical applications unclear. Although the theoretical section provides a detailed analysis of bias characteristics between cluster randomized design and other methods, there is no empirical validation, particularly in terms of bias performance and computational complexity across different scenarios.

Certain technical details regarding experimental parameters are not fully explained, making it difficult to thoroughly evaluate the reproducibility and transparency of the results. For example, the simulation experiments lack an analysis of the robustness of key parameter choices (e.g., the number of users and items) and their impact on the results.

While the paper mentions that bias is influenced by network weights and cluster quality, it does not deeply analyze the sensitivity of these parameters on the outcomes, which may restrict the applicability of the conclusions.

The paper's formulas lack sufficient introduction and explanation, and there are no formula numbers provided for easy reference.

**Questions:**

How do the authors validate the assumptions regarding the weight distribution of user interaction networks? If this assumption does not hold, how would it affect the predictive ability of the model?

Why does the empirical section not include a direct performance comparison between the cluster randomized design and other existing methods?

**Reviewer Confidence:**

3: The reviewer is confident but not certain that the evaluation is correct

**Scope:**

3: The work is somewhat relevant to the Web and to the track, and is of narrow interest to a sub-community

---

### Official Review · Reviewer_qjbA · 2024-12-02

**Novelty:** 4
**Technical Quality:** 4

**Review:**

## Summary

This paper investigates "symbiosis bias," a novel challenge in A/B testing of recommendation algorithms arising from shared training data between algorithms under comparison. It critiques how such bias impacts Total Treatment Effect (TTE) estimates, violating the stable unit treatment value assumption (SUTVA). The authors introduce three experimental designs—cluster-randomized, data-diverted, and user-corpus co-diverted experiments—to mitigate this bias. Using theoretical models, simulations, and real-world experiments on a large-scale recommender system, the paper evaluates these designs' efficacy. The findings highlight that while each approach reduces symbiosis bias, they introduce trade-offs such as reduced training data or operational challenges.

## Strengths
- The identification and detailed analysis of symbiosis bias address an overlooked but significant issue in A/B testing for recommendation systems.
- The paper proposes three experimental methods to tackle symbiosis bias, expanding the existing toolbox for practitioners.
- The paper makes a strong combination of theoretical modeling, simulations, and validation using real-world data strengthens the study's conclusions.

## Weaknesses
- The computational and operational feasibility of deploying the proposed designs in large-scale dynamic settings is underexplored.
- The paper mentions trade-offs (e.g., reduced training data) but provides limited guidance on mitigating these drawbacks.
- Certain theoretical assumptions (e.g., approximating user-item networks) may not fully capture real-world complexities, limiting applicability in heterogeneous environments.

**Questions:**

- How do the proposed experimental designs scale computationally for larger datasets or more complex recommendation systems?
- Could you provide additional insights into mitigating the drawbacks (e.g., data inefficiency) of the data-diverted and user-corpus co-diverted designs?
- How sensitive are the findings to the quality of user clusters in the cluster-randomized experiments? Could clustering be improved dynamically?

**Reviewer Confidence:**

1: The reviewer's evaluation is an educated guess

**Scope:**

4: The work is relevant to the Web and to the track, and is of broad interest to the community

---

### Official Review · Reviewer_g7UN · 2024-12-02

**Novelty:** 5
**Technical Quality:** 5

**Review:**

strengths
This work proposes symbiosis bias and theoretically defines how the bias is computed in three experiment settings: user clustering, data diversion, and user-corpus co-diversion.
- The effectiveness of mitigating symbiosis bias is validated in simulation experiments.
- The existence of symbiosis bias is empirically proved in real-world A/B test data.

weakness
- This work has not provided experiments on real-world data for data-diversion and user-corpus co-diversion.
- The data diversion design needs to be more straightforward. How is it expressed in Figure 1?

**Questions:**

It would be better to provide experiments on real-world data for data-diversion and user-corpus co-diversion and visualize the results.

**Reviewer Confidence:**

3: The reviewer is confident but not certain that the evaluation is correct

**Scope:**

4: The work is relevant to the Web and to the track, and is of broad interest to the community

---

### Official Review · Reviewer_HbG1 · 2024-12-03

**Novelty:** 6
**Technical Quality:** 6

**Review:**

### Quality
The article "Reducing Symbiosis Bias Through Better A/B Tests of Recommendation Algorithms" presents a compelling examination of biases inherent in algorithmic comparisons, especially symbiosis bias. It methodically explores three experimental designs and validates findings through simulations and real-world data, showcasing a rigorous scientific approach. The technical quality is strong, underpinned by clear theoretical frameworks and empirical evidence.

### Clarity
The writing is generally clear, though sections delving into mathematical modeling may be challenging for readers without a background in statistics or causal inference. The inclusion of simulations and practical examples aids comprehension but could benefit from additional simplification or graphical summaries to make the findings more accessible.

### Originality
The concept of "symbiosis bias" is novel, representing a significant contribution to the field of recommendation systems. The work stands out for introducing and comparing three experimental designs, one of which (cluster randomization) had not been previously explored in this context.

### Significance
This work has practical implications for researchers and industry practitioners aiming to reduce biases in A/B testing of recommendation algorithms. By identifying limitations of traditional methods and proposing alternatives, it fills a critical gap in the literature.

### Pros
- Novelty: Introduces and defines symbiosis bias in algorithm evaluation.
- Methodological Rigor: Combines theoretical modeling, simulation, and real-world data.
- Practical Implications: Offers actionable insights for experiment design in industrial settings.
- Comprehensive Coverage: Evaluates multiple approaches to mitigate symbiosis bias.

### Cons
- Cluster Randomization Limitations: Efficacy highly depends on the quality of clusters, which might not be achievable in many practical scenarios.
- Unaddressed Variance Issues: Variance implications of the experimental designs are briefly discussed but lack detailed exploration.

**Questions:**

- The discussion on variance suggests wider confidence intervals in certain designs. Could this make such designs impractical for large-scale systems?
- What practical challenges or trade-offs did you observe in applying cluster randomization in real-world systems, and how do you propose to address them?
- Do you plan to validate the results on datasets with different characteristics or from other industries to enhance the generalizability of your findings?

**Reviewer Confidence:**

3: The reviewer is confident but not certain that the evaluation is correct

**Scope:**

3: The work is somewhat relevant to the Web and to the track, and is of narrow interest to a sub-community